# Clinical and pathological implications of the presence of MECA-79-expressing tumor cells in pathological stage IA lung adenocarcinoma

Tomohito Saito[1]*, Mitsuaki Ishida[2], Tomoya O. Akama[3], Shiho Hattori[1], Natsumi Maru[1], Takahiro Utsumi[1], Aki K. Kobayashi[1], Kento J. Fukumoto[1], Hiroshi Matsui[1], Yohei Taniguchi[1], Yoshinobu Hirose[2], Katsuyasu Kouda[4], Tomohiro Murakawa[1]

1 Department of Thoracic Surgery, Kansai Medical University, Osaka, Japan, 2 Department of Pathology, Osaka Medical and Pharmaceutical University, Osaka, Japan, 3 Department of Pharmacology, Kansai Medical University, Osaka, Japan, 4 Department of Hygiene and Public Health, Kansai Medical University, Osaka, Japan

* saitotom@hirakata.kmu.ac.jp

## Abstract

Approximately 15% of patients with resected pathological stage IA lung adenocarcinoma develop recurrent disease, indicating the formation of a cancer metastasis-promoting microenvironment, and highlighting the importance of identifying early prognostic biomarkers. The MECA-79 epitope is a glycan structure modulating immune response, normally expressed on high endothelial venules. Ectopic MECA-79 expression has been recently reported in several cancer cells and is associated with poor prognosis. In this retrospective cohort study, we aimed to investigate the clinical and pathological significance of tumoral MECA-79 expression in early-stage lung cancer. Immunohistochemical analysis for MECA-79 was performed in 195 patients with pathological stage IA lung adenocarcinoma undergoing lobectomy. Clinical, radiological, and pathological factors were assessed, and recurrence-free survival (RFS) was analyzed using Kaplan–Meier analysis and univariable Cox regression proportional hazards models. Multivariable Cox analyses were performed as exploratory analyses only due to the limited number of recurrence events. Tumoral MECA-79 expression was observed in 5.1% of cases (n = 10). Patients with MECA-79⁺ tumor cells exhibited a larger pathological invasive size (2.1 vs. 1.6 cm, $P = 0.044$), higher rates of vascular invasion (90.0% vs. 40.0%, $P = 0.0023$), and increased 5-year postoperative recurrence (40.0% vs. 7.6%, $P = 0.0061$). Kaplan–Meier analysis demonstrated significantly worse RFS for patients with MECA-79⁺ tumor cells (5-year rate: 54.9% vs. 87.4%, $P = 0.003$). The univariate Cox regression model identified body mass index, histological grade based on the International Association for the Study of Lung Cancer histological grading system, vascular invasion, spread through air spaces, and the presence of MECA-79⁺ tumor cells as prognostic factors. Our results indicate that tumoral MECA-79 expression is associated with the

**Data availability statement:** All relevant data are within the manuscript and Supporting Information files includes relevant data of 195 pStage IA LUAD cases analyzed for this study.

**Funding:** JSPS KAKENHI Grant # JP20K09184 and # JP24K12309.

**Competing interests:** The authors have declared that no competing interests exist.

recurrence of resected pathological stage IA lung adenocarcinoma; however, these findings should be validated in multicenter, stage-matched cohorts.

## Introduction

Lung cancer is the leading cause of cancer-related deaths worldwide, with an estimated 1.8 million lung cancer-related deaths in 2022 [1]. Anatomic surgical resection is the mainstay definitive treatment for early-stage non-small cell lung cancer (NSCLC), and pathological stage IA disease accounts for approximately 50% of resectable NSCLC [2]. However, postoperative recurrence rate of pathological stage IA NSCLC remains as high as approximately 15% [3]. Our primary clinical question is why such a high proportion of patients with pathological stage IA NSCLC experience postoperative recurrence despite complete resection. Delineation of the underlying mechanisms of recurrence is essential for improving clinical outcomes.

Sialyl Lewis$^X$ (sLe$^X$), an established serum tumor marker, is an important glycan epitope in tumor progression through selectin interactions. A sulfated derivative of sLe$^X$, 6-sulfo sLe$^X$, is also an important L-, P-, and E-selectin ligand. Immunohistochemically, 6-sulfo sLe$^X$ is recognized by MECA-79 antibody exclusively when 6-sulfo sLe$^X$ is present on a particular glycan core structure, the MECA-79 epitope: 6-sulfo $N$-acetyllactosamine on the extended core 1 $O$-glycans, Galβ1→4(sulfo→6)GlcNAcβ1→3Galβ1→GalNAc→$O$-R [4]. In contrast to sLe$^X$, 6-sulfo sLe$^X$ as well as MECA-79 epitope have mainly been studied on non-malignant cells such as high endothelial venules in an immune modulation context. Recently, ectopic MECA-79 epitope expression in human cancer cells has been discovered and proven to be associated with poor prognosis in cholangiocarcinoma as well as gastric, bladder, and breast cancers [5–10]. However, to date, the presence of MECA-79$^+$ tumor cells in NSCLC and the clinical and pathological significance of such presence have not been described.

The present study aimed to investigate the clinical and pathological significance of the presence of MECA-79$^+$ tumor cells in pathological stage IA lung adenocarcinoma (LUAD).

## Materials and methods

This study was conducted in accordance with the principles outlined in the Declaration of Helsinki as revised in 2013. This study was approved by the Kansai Medical University Hospital Research Ethics Committee (approval number: 2016662; approval date: September 12, 2016). The requirement for informed consent was waived as this retrospective study analyzed anonymized data and posed minimal risk to participants. Chart reviews were conducted on August 26, 2021, and December 31, 2024, for research purposes. At no point after data collection did the authors have access to information that could identify individual participants. Archived pathological samples were collected and anonymized prior to analysis, and immunostaining and evaluation of these samples were performed between August 26, 2021, and October 29, 2021.

## Patient selection

A flow diagram illustrating the patient selection process is shown in Fig 1. The records of 1014 patients who underwent surgery for lung cancer at Kansai Medical University Hospital between January 1, 2009, and December 31, 2017, were reviewed. The inclusion criteria were a pathological diagnosis of primary invasive LUAD, pathological stage IA (as per the 9th edition of the TNM Classification system) [11], complete resection (R0) with lobectomy or greater, and tissue available for this study.

The rationale for our inclusion criteria was as follows: We specifically analyzed pathological stage IA cases to minimize confounding variables associated with advanced disease stages. Furthermore, we restricted our cohort to adenocarcinoma to eliminate potential confounding effects resulting from histological variations, given the known heterogeneity in biological behavior among NSCLC subtypes [12]. We only included the patients who underwent complete (R0) resection with lobectomy to avoid confounding effects related to surgical procedures or residual disease status.

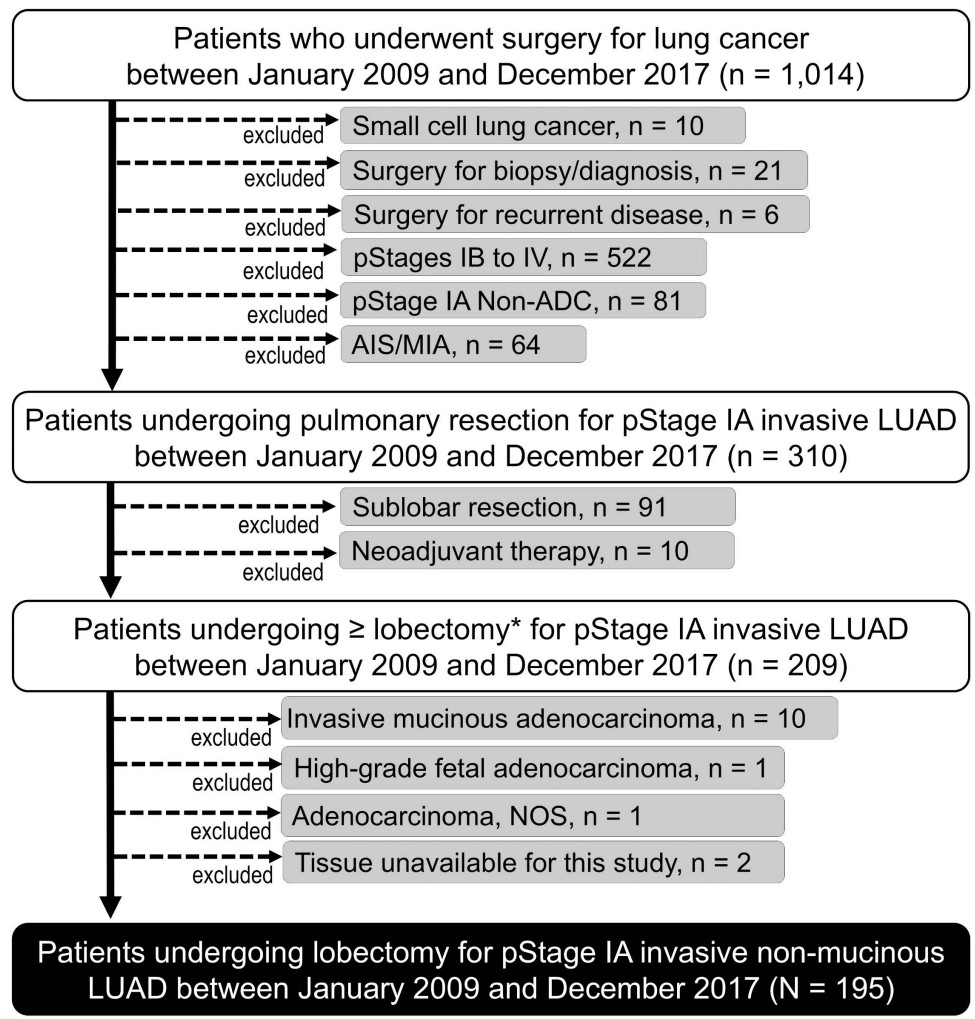

**Fig 1. Flow diagram of the patient selection process.** Briefly, out of 1014 patients undergoing surgery for lung cancer at Kansai Medical University Hospital between January 1, 2009, and January 31, 2017, a total of 195 patients who underwent complete resection with lobectomy for pathological stage IA invasive adenocarcinoma of the lung with no neoadjuvant therapy were included in this study. Adenocarcinoma *in situ* and minimally invasive adenocarcinoma were excluded.

The exclusion criteria were adenocarcinoma *in situ*, minimally invasive adenocarcinoma, surgery for biopsy/diagnosis, and patients who had received neoadjuvant therapy. Based on these criteria, 195 patients who had undergone pulmonary lobectomy with lobe-specific lymph node dissection were included for analysis in this study.

## Diagnosis of invasive LUAD

The resected specimens were handled in accordance with the Japan Lung Cancer Society General Rule for Clinical and Pathological Records of Lung Cancer [13]. The pathological diagnosis of invasive LUAD was made in accordance with the 2015 World Health Organization histological classification of lung tumors [14]. Grading was determined according to The International Association for Study of Lung Cancer (IASLC) grading system for invasive LUAD based on a recent report [15]. In addition to vascular invasion and lymphatic permeation, spread through air spaces (STAS) was evaluated as previously described [16].

## Immunohistochemical analyses

Formalin-fixed, paraffin-embedded tissues cut into 5-µm-thick sections were used for hematoxylin-eosin staining and for immunohistochemical analysis using MECA-79 antibody (1:100; MECA-79; Novus Biologicals, Englewood, CO, USA). Tumoral expression of MECA-79 was reviewed by a trained pathologist (MI). Representative microscopic images of MECA-79[+] tumor cells in this study are shown in Fig 2.

## Radiological features on chest computed tomography and 18-fluorodeoxyglucose positron emission tomography

Preoperative chest computed tomography (CT) images were obtained with a slice thickness of 1–10 mm and a field of view of 14–39 cm. The type of pulmonary nodule (solid, part-solid, or pure ground-glass nodule) and the size of the nodule (total size and solid size) were determined only in patients who underwent high-resolution CT with a slice thickness of 1–2 mm, in accordance with previous reports [17,18]. For patients who had undergone preoperative 18-fluorodeoxyglucose positron emission tomography/CT (FDG-PET/CT) and whose radiological diameter of the pulmonary nodule was ≥ 1.0 cm, the maximum standardized uptake value within the primary lesion was recorded.

## Analysis using publicly available genomic databases

For a comprehensive examination of gene expression patterns focusing on the mechanistic background of tumoral MECA-79 expression, a robust, publicly available cancer genomics resource was used. The LUAD dataset from The Cancer Genome Atlas (TCGA), which includes 565 cases and is a widely used resource for cancer genomics, was accessed via the cBio Cancer Genomics Portal (https://www.cbioportal.org/) [19]. The cBio Cancer Genomics Portal was used to

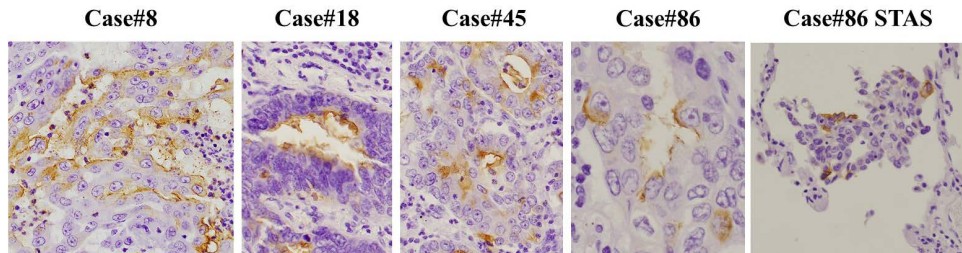

**Fig 2. Representative microscopic images of tumoral MECA-79 expression in human lung adenocarcinoma cells.** Tumoral expression of MECA-79 was observed predominantly in the cell membrane and occasionally in the cytoplasm. In case #86, MECA-79[+] tumor cells were also found in cell clusters of spread through air spaces (STAS). MECA-79 antibody (Novus Biologicals, Englewood, CO, USA) was used for immunostaining.

analyze gene expression in LUAD tumor cells. Specifically, we examined the expression levels of genes associated with MECA-79 antigen biosynthesis (e.g., *B3GNT3*, *B4GALT4*, *CHST2*, and *CHST4*) [20–23]. Gene expression levels in LUAD tumor cells were compared with those of normal cells within TCGA LUAD dataset.

### Statistical analysis

The Mann–Whitney U test was used to analyze continuous variables, while Fisher's exact test was used for categorical variables in comparing two groups. Postoperative recurrence of LUAD was diagnosed considering the evidence provided by physical examination and diagnostic imaging such as CT, magnetic resonance imaging (MRI), and FDG-PET/CT as previously described [24]. Recurrent disease was pathologically confirmed when clinically feasible. The Kaplan–Meier estimation curve was used to visually analyze OS and recurrence-free survival (RFS), and the difference between groups was evaluated using the log-rank test. OS was calculated as the interval between the date of pulmonary resection and the date of all-cause death. RFS was calculated as the interval between the date of pulmonary resection and the date of first documented recurrence or all-cause death. The date of the last follow-up was used for censored patients.

Univariable Cox proportional hazards regression was primarily used to evaluate the association between clinical and pathological characteristics with recurrence-free survival (RFS) in patients with LUAD. Factors with statistically significance ($P < 0.05$) in the univariable analysis were considered candidate risk factors. To minimize overfitting given the limited number of recurrence or death events (n = 30), multivariable Cox models–restricted to two candidate variables, one of which was the presence of MECA-79⁺ tumor cells–were prespecified as exploratory analyses and are presented in the Supplementary Material (S1 Table).

Data were analyzed using JMP Pro version 18.2.0 (JMP Statistical Discovery LLC, NC, USA) for Windows (Microsoft, Redmond, WA, USA). The Kaplan–Meier estimation curve was visualized using Prism version 6.0.7 (GraphPad Software, MA, USA). Statistical significance was set at $P < 0.05$.

## Results

The clinical and pathological characteristics of the study population according to the presence or absence of MECA-79⁺ tumor cells are summarized in Table 1. Of the 195 patients included for analysis, ten (5.1%) showed MECA-79⁺ tumor cells. Radiological solid and pathological invasive sizes were slightly greater in patients with MECA-79⁺ tumor cells than in their counterparts ($P = 0.046$ and $0.044$, respectively). Vascular invasion and recurrence within five postoperative years were more commonly observed in patients with MECA-79⁺ tumor cells ($P = 0.0023$ and $0.0061$, respectively). The localization of postoperative recurrence was not significantly different between the two study groups ($P = 0.77$).

FDG-PET/CT, fluorodeoxyglucose-positron emission tomography/computed tomography; HRCT, high-resolution computed tomography; IASLC, International Association for the Study of Lung Cancer; SUVmax, maximum standardized uptake value.

While OS showed no significant difference between the two study groups ($P = 0.32$, Fig 3A), RFS was significantly worse for patients with MECA-79⁺ tumor cells than for those with no MECA-79⁺ tumor cells ($P = 0.003$, Fig 3B).

Additionally, univariate Cox proportional hazards regression analysis identified body mass index (hazard ratio [HR] = 0.86, confidence interval [CI]: 0.76–0.96, $P = 0.009$), IASLC grade of LUAD (HR = 2.23, 95% CI: 1.05–4.73, $P = 0.036$), vascular invasion (HR = 3.31, 95% CI: 1.51–7.25, $P = 0.003$), STAS (HR = 4.34, 95% CI: 2.11–8.91, $P < 0.001$), and presence of MECA-79⁺ tumor cells (HR = 4.38, 95% 1.51–12.7, $P = 0.007$) as candidate prognostic factors of RFS (Table 2).

Additionally, results of exploratory multivariable Cox proportional hazards regression analyses are provided in the Supplementary Material (S1 Table).

**Table 1. Baseline clinical and pathological characteristics of patients undergoing lobectomy for pathological stage IA lung adenocarcinoma according to tumoral MECA-79 status (n = 195).**

| *Characteristics* | MECA-79–expressing tumor cells | | *P-value* |
| --- | --- | --- | --- |
| | **Present (n = 10)** | **Absent (n = 185)** | |
| **Age at the time of surgery, years** | 69 [43–79] | 69 [38–84] | 0.63 |
| *Sex* | | | |
| **Male** | 8 (80.0%) | 96 (51.9%) | 0.11 |
| **Female** | 2 (20.0%) | 89 (48.1%) | |
| *Smoking history* | | | |
| **Never smoked** | 2 (20.0%) | 75/181 (41.4%) | 0.32 |
| **Ever smoked** | 8 (80.0%) | 106/181 (58.6%) | |
| **Body mass index** | 20.7 [16.2–26.3] | 22.7 [15.0–32.6] | 0.27 |
| **Carcinoembryonic antigen level, ng/mL** | 3.8 [2.0–14.5] | 2.6 [1.0–34.9] | 0.074 |
| *Radiological size on HRCT* | | | |
| **Total size, cm** | 2.0 [1.7–3.7] (n = 6) | 2.0 [0.7–6.3] (n = 128) | 0.51 |
| **Solid size, cm** | 2.0 [1.7–2.8] (n = 6) | 1.6 [0–4.0] (n = 128) | 0.046 |
| *Radiological pattern on HRCT* | | | |
| **Solid nodule** | 5/6 (83.3%) | 59/128 (46.1%) | 0.22 |
| **Part-solid nodule** | 1/6 (16.7%) | 66/128 (51.6%) | |
| **Pure ground-glass nodule** | 0 | 3/128 (2.3%) | |
| **SUVmax in $^{18}$F-FDG-PET/CT** | 3.8 [0–8.6] (n = 9) | 2.4 [0–9.6] (n = 142) | 0.12 |
| *Pathological stage** | | | |
| **pStage IA1** | 0 | 32 (17.3%) | 0.20 |
| **pStage IA2** | 5 (50.0%) | 103 (55.7%) | |
| **pStage IA3** | 5 (50.0%) | 50 (27.0%) | |
| **Pathological invasive size, cm** | 2.1 [1.2–2.6] | 1.6 [0.5–3.0] | 0.044 |
| *IASLC grade of lung adenocarcinoma* | | | |
| **Grade 1** | 0 | 33 (17.8%) | 0.19 |
| **Grade 2** | 6 (60.0%) | 114 (61.6%) | |
| **Grade 3** | 4 (40.0%) | 38 (20.5%) | |
| *Vascular invasion* | | | |
| **Present** | 9 (90.0%) | 74 (40.0%) | 0.0023 |
| **Absent** | 1 (10.0%) | 111 (60.0%) | |
| *Lymphatic permeation* | | | 0.50 |
| **Present** | 8 (80.0%) | 117 (63.2%) | |
| **Absent** | 2 (20.0%) | 68 (36.8%) | |
| *Spread through air spaces* | | | 0.42 |
| **Present** | 3 (30.0%) | 36 (19.5%) | |
| **Absent** | 7 (70.0%) | 149 (80.5%) | |
| *Recurrence within 5 years after surgery* | | | |
| **Observed** | 4 (40.0%) | 13 (7.0%) | 0.0061 |
| **Not observed** | 6 (60.0%) | 172 (93.0%) | |
| *Localization of postoperative recurrence* | | | |
| **Locoregional-only** | 3/4 (75.0%) | 6/13 (46.2%) | 0.77 |
| **Locoregional and distant** | 1/4 (25.0%) | 4/13 (30.8%) | |
| **Distant-only** | 0 | 3/13 (23.1%) | |
| **Duration of follow-up, years** | 5.3 [2.3–12.8] | 5.6 [0.6–13.6] | 0.97 |

*Pathological stage as per the 9[th] edition of the TNM Classification system. Values for continuous variables are provided as the median with range.

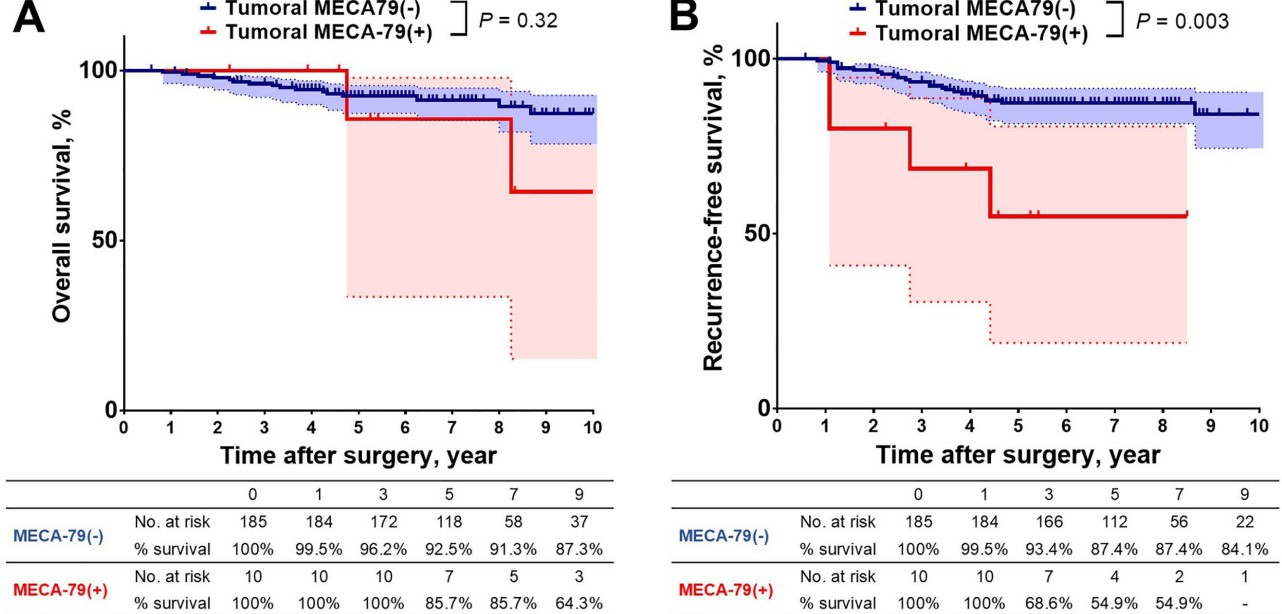

**Fig 3. Overall and recurrence-free survival according to tumoral MECA-79 expression status. (A)** Overall survival was not significantly different regardless of tumoral MECA-79 expression status. **(B)** Recurrence-free survival was significantly worse for patients with MECA-79⁺ tumor cells than for those with no MECA-79⁺ tumor cells (5-year recurrence-free survival, 54.9% vs. 87.4%, respectively, *P*<0.001).

Table 2. Univariate Cox proportional hazards regression analysis of clinical and pathological characteristics for predicting recurrence-free survival.

| Characteristics | Univariate Cox proportional hazards regression model | | | Overall P-value |
| --- | --- | --- | --- | --- |
| | HR | 95% CI | *P*-value | |
| Age | 1.04 | 0.99–1.09 | 0.12 | |
| Sex: Male vs. female | 2.16 | 0.99–4.72 | 0.053 | |
| Smoking status: ever-smokers | 2.31 | 0.99–5.41 | 0.053 | |
| Body mass index | 0.86 | 0.76–0.96 | 0.009 | |
| Carcinoembryonic antigen level, ng/mL | 1.03 | 0.96–1.08 | 0.40 | |
| *Pathological stage | | | | 0.25 |
| IA2 vs. IA1 | 2.76 | 0.64–11.9 | 0.17 | |
| IA3 vs. IA1 | 2.93 | 0.64–13.4 | 0.17 | |
| IASLC grade of LUAD: 3 vs. 1–2 | 2.23 | 1.05–4.73 | 0.036 | |
| Vascular invasion: (+) vs. (-) | 3.31 | 1.51–7.25 | 0.003 | |
| Lymphatic permeation: (+) vs. (-) | 2.61 | 0.99–6.87 | 0.052 | |
| STAS: (+) vs. (-) | 4.34 | 2.11–8.91 | <0.001 | |
| MECA-79⁺tumor cells: (+) vs. (-) | 4.38 | 1.51–12.7 | 0.007 | |

HR, hazard ratio; CI, confidence interval; IASLC, International Association for the Study of Lung Cancer; LUAD, lung adenocarcinoma; STAS, spread through air spaces.

*Pathological stage as per the ninth edition of the TNM classification system.

## Discussion

Our retrospective immunohistochemical analysis of 195 resected pathological stage IA LUAD cases indicated that the presence of MECA-79[+] tumor cells might be associated with postoperative recurrence within 5 years after surgery and with vascular invasion, but neither with lymphatic permeation nor STAS.

To date, there have been six reports on tumoral expression of the MECA-79 epitope in human cancer: gastric cancer [5], cholangiocarcinoma [6–8], bladder cancer [9], and breast cancer [10]. The proportion of patients showing MECA-79[+] tumor cells seem to vary by cancer type: 28% for gastric cancer [5], 42%–70% for cholangiocarcinoma [6,8], 20% for bladder cancer [9], and 0% for hepatocellular carcinoma [6]. In gastric cancer, the presence of MECA-79[+] tumor cells seems to be associated with venous invasion, distant metastasis, and poor cancer-specific survival in gastric cancer. In our study, we observed a relatively low MECA-79[+] tumor-cell positivity rate (5.1%) and associations of the presence of MECA-79[+] tumor cells with vascular invasion and postoperative recurrence in LUAD, while we did not find a significant increase in distant metastasis in patients with MECA-79[+] tumor cells.

The MECA-79 biosynthetic pathway involves a series of glycosylation and sulfation steps on proteins that generate specialized O-glycan structures on proteins (Fig. 4A). Specifically, the enzymes beta-1,3-N-acetylglucosamine transferase 3 (B3GNT3) [20], carbohydrate sulfotransferase (CHST) family (CHST2 and CHST4) [21,22], and beta-1,4-galactosyltransferase 4 (B4GALT4) [23] have been shown to be essential for the production of MECA-79 antigen.

In an analysis of TCGA LUAD dataset (n = 565) using Bio Cancer Genomics Portal (https://www.cbioportal.org/) [19], 87.5% of human LUAD samples showed gene alterations in B3GNT3, B4GALT4, CHST2, or CHST4, predominantly mRNA upregulation (78.6%), as shown for B3GNT3 and B4GALT4 in Fig. 4B. While such upregulation may provide a biological background for MECA-79 biosynthesis, MECA-79 epitope expression was rare (5.1%) in our cohort, despite frequent mRNA upregulation in TCGA. This discrepancy highlights a well-recognized phenomenon in glycobiology:

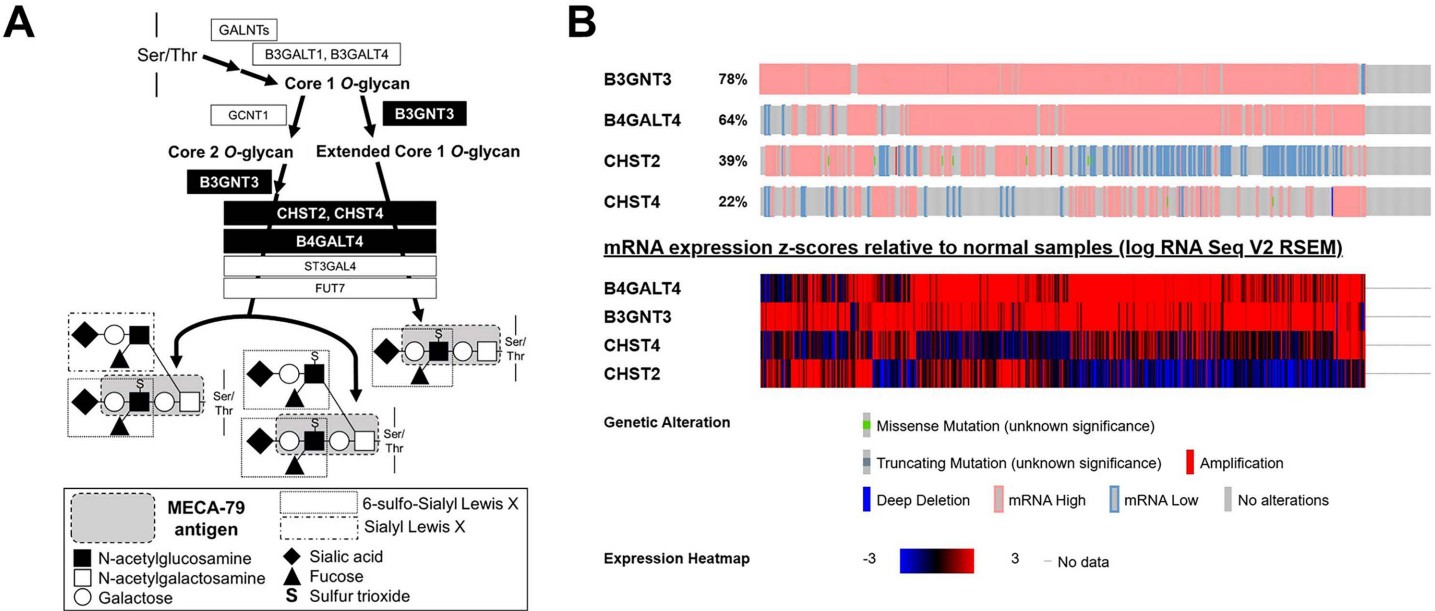

**Fig. 4. Overview of MECA-79 antigen biosynthesis and heatmap of gene expressions associated with MECA-79 production.** (A) Overview of biosynthesis of the MECA-79 antigen. (B) Heatmap of mRNA expression z-scores relative to normal samples in representative genes associated with MECA-79 antigen biosynthesis. Specifically, B3GNT3 and B4GALT4 showed remarkable upregulation. Genes associated with MECA-79 were obtained from TCGA.

mRNA levels of glycosylation enzymes do not necessarily correspond to the presence of glycan epitopes due to post-transcriptional regulation and the requirement of carrier proteins [25,26]. These findings suggest that TCGA data should be regarded as contextual mechanistic background rather than prognostic evidence. Nevertheless, future studies incorporating glycoproteomic profiling are needed to resolve this gap.

The pathophysiological mechanism linking tumoral MECA-79 expression and cancer recurrence remains to be fully elucidated. Given that the MECA-79 epitope comprises a part of 6-sulfo sLe$^X$, the pro-metastatic roles of tumoral 6-sulfo sLe$^X$ expression could explain the increase in vascular invasion and cancer recurrence observed in the presented study. Tumoral 6-sulfo sLe$^X$ has been shown to bind E-selectin on vascular endothelial cells via calcium-dependent interactions, facilitating vascular invasion in bladder cancer [9]. Further, tumoral 6-sulfo sLe$^X$ expression enables cancer cells to form microthrombi with platelets by binding to P-selectin, which may create protective niches that evade immunological surveillance [27]. However, the anti-metastatic role of tumoral 6-sulfo sLe$^X$ has also been shown in oral squamous cell carcinoma [28]. Tumoral 6-sulfo sLe$^X$ can attract CD8 + cytotoxic T lymphocytes, promoting the formation of immune synapses. Another possibility could be that tumoral MECA-79 expression is phenotypic rather than functional and that the underlying mechanism of upregulation of MECA-79 in cancer cells promotes cancer recurrence. For example, analysis of TCGA LUAD dataset using the cBio Cancer Genomics Portal revealed that *B3GNT3* is co-expressed with multiple genes associated with poor prognosis in patients with LUAD (Table 3) [29–38]. This finding suggests that MECA-79$^+$ LUAD tumor cells may enhance their malignant potential through the upregulation of these co-expressed genes.

The limitations of the presented study were as follows: First, our results are based on a retrospective single-institutional study with a small sample size, particularly the MECA-79$^+$ group (n = 10). This imbalance increases statistical fragility, limits power, and raises the possibility of collinearity with vascular invasion, which may confound the apparent prognostic effect of MECA-79. Therefore, the present study should be regarded as exploratory, and validation in a prospective, multi-institutional study with a larger sample size—ideally including stratification by vascular invasion—is required. Second, this study only included pathological stage IA LUAD. Thus, the clinical and pathological significance of the presence of MECA-79$^+$ tumor cells in patients with more advanced pathological stages (i.e., IB–IV) or with other histological types, such as non-adenocarcinoma NSCLC (e.g., squamous cell carcinoma), should be investigated in future studies. Third, we did not have background data on gene alterations such as epidermal growth factor receptor, Kirsten rat sarcoma

**Table 3. Co-expression genes correlated with *B3GNT3* or *B4GALT4* in The Cancer Genome Atlas lung adenocarcinoma samples.**

| | Co-expression gene | Spearman's correlation | P-value | q-value | Prognostic implication in LUAD |
|---|---|---|---|---|---|
| **B3GNT3** | GJB3 | 0.631 | 6.63e-58 | 1.33e-53 | Poor [Ref. 29] |
| | KRT19 | 0.588 | 8.25e-49 | 8.26e-45 | Poor [Ref. 30] |
| | ITGB4 | 0.581 | 2.28e-47 | 1.52e-43 | Poor [Ref. 31] |
| | LAMC2 | 0.556 | 1.16e-42 | 5.78e-39 | Poor [Ref. 32] |
| | PTPRH | 0.548 | 2.77e-41 | 1.11e-37 | Poor [Ref. 33] |
| | PRSS22 | 0.531 | 2.11e-38 | 7.04e-35 | N/A |
| | KRT16 | 0.529 | 3.62e-38 | 1.03e-34 | Poor [Ref. 34] |
| | PLIN3 | 0.521 | 7.77e-37 | 1.94e-33 | Poor [Ref. 35] |
| | C19ORF33 | 0.520 | 9.49e-37 | 2.11e-33 | N/A |
| | CRYBG2 | 0.511 | 2.59e-35 | 5.19e-32 | N/A |
| | GJB4 | 0.509 | 5.07e-35 | 8.78e-32 | Poor [Ref. 36] |
| | KRT8 | 0.509 | 5.27e-35 | 8.78e-32 | Poor [Ref. 37] |
| **B4GALT4** | TFG | 0.560 | 1.60e-43 | 3.19e-39 | N/A |
| | KPNA1 | 0.554 | 2.76e-42 | 2.76e-38 | N/A |
| | COPB2 | 0.520 | 1.02e-36 | 6.82e-33 | Poor [Ref. 38] |
| | ISGF11 | 0.518 | 2.28e-36 | 1.14e-32 | N/A |

viral oncogene homolog, and anaplastic lymphoma kinase mutations. Thus, the relationship between tumoral MECA-79 expression and the genetic background will need to be clarified in future studies. Fourth, TCGA analysis revealed frequent upregulation of biosynthetic enzymes despite rare MECA-79 epitope expression in our cohort. This discrepancy reflects a common disconnect between transcript levels and glycan presentation, underscoring that TCGA findings cannot be considered supportive prognostic evidence but only contextual background.

In conclusion, our results indicate that the presence of MECA-79[+] tumor cells could be associated with vascular invasion and postoperative recurrence of resected pathological stage IA LUAD. Further investigations are necessary to validate our results and delineate the underlying mechanisms.

## Supporting information

**S1 Table. Multivariate Cox proportional hazards models including two\* candidate factors for recurrence-free survival.**
(DOCX)

**S2 File. Raw data of 195 pStage IA LUAD cases for analysis.**
(XLSX)

## Acknowledgments

The authors would also like to thank Editage (www.editage.com) for English language editing.

## Author contributions

**Conceptualization:** Tomohito Saito.

**Data curation:** Tomohito Saito, Mitsuaki Ishida, Shiho Hattori, Natsumi Maru, Takahiro Utsumi, Kento J. Fukumoto, Hiroshi Matsui, Yohei Taniguchi, Katsuyasu Kouda.

**Formal analysis:** Tomohito Saito, Katsuyasu Kouda.

**Funding acquisition:** Tomohito Saito.

**Investigation:** Tomohito Saito.

**Methodology:** Tomohito Saito, Mitsuaki Ishida, Tomoya O. Akama, Katsuyasu Kouda.

**Project administration:** Tomohito Saito, Shiho Hattori.

**Resources:** Tomohito Saito, Mitsuaki Ishida, Tomoya O. Akama, Takahiro Utsumi, Kento J. Fukumoto, Yohei Taniguchi.

**Software:** Natsumi Maru, Takahiro Utsumi, Kento J. Fukumoto, Yohei Taniguchi.

**Supervision:** Mitsuaki Ishida, Tomoya O. Akama, Yoshinobu Hirose, Katsuyasu Kouda, Tomohiro Murakawa.

**Validation:** Tomohito Saito, Yoshinobu Hirose, Katsuyasu Kouda.

**Visualization:** Tomohito Saito.

**Writing – original draft:** Tomohito Saito.

**Writing – review & editing:** Tomohito Saito, Mitsuaki Ishida, Natsumi Maru, Takahiro Utsumi, Aki K. Kobayashi, Kento J. Fukumoto, Hiroshi Matsui, Yohei Taniguchi, Katsuyasu Kouda, Tomohiro Murakawa.

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
