## [Decision Letter · Decision Letter 0]

28 May 2025

PONE-D-25-16991Clinical and pathological implications of the presence of MECA-79-expressing tumor cells in pathological stage IA lung adenocarcinomaPLOS ONE

Dear Dr. Saito,

Thank you for submitting your manuscript to PLOS ONE. After careful consideration, we feel that it has merit but does not fully meet PLOS ONE’s publication criteria as it currently stands. Therefore, we invite you to submit a revised version of the manuscript that addresses the points raised during the review process.

The reviewers provided constructive feedback, particularly regarding the statistical analysis, including the use of univariable and multivariable models given the limited number of events. Please incorporate more rigorous statistical methods to avoid potential overfitting and strengthen the validity of the findings. Furthermore, integrating a bioinformatic analysis of publicly available datasets to explore the expression of genes involved in MECA-79 glycan biosynthesis could provide additional mechanistic insights and further support the clinical relevance of MECA-79 in tumor biology.

We look forward to receiving your revised manuscript.

Kind regards,

Hyun-Sung Lee, M.D., Ph.D.

Academic Editor

PLOS ONE

Additional Editor Comments (if provided):

Reviewers' comments:

Reviewer's Responses to Questions

**Comments to the Author**

1. Is the manuscript technically sound, and do the data support the conclusions?

Reviewer #1: No

Reviewer #2: Yes

Reviewer #3: Yes

2. Has the statistical analysis been performed appropriately and rigorously? 

Reviewer #1: No

Reviewer #2: Yes

Reviewer #3: No

3. Have the authors made all data underlying the findings in their manuscript fully available?

Reviewer #1: Yes

Reviewer #2: Yes

Reviewer #3: Yes

4. Is the manuscript presented in an intelligible fashion and written in standard English?

Reviewer #1: No

Reviewer #2: Yes

Reviewer #3: Yes

5. Review Comments to the Author

Reviewer #1: It was a great pleasure to review the manuscript “Clinical and pathological implications of the presence of MECA-79-expressing tumor cells in pathological stage ⅠA lung adenocarcinoma” by Tomohito Saito, et al. In this study, the authors investigated the relationship between MECA-79(+) tumors and prognosis in 195 patients who underwent lobectomy for stage 1A lung adenocarcinoma. They revealed that the solid size and invasive size of MECA-79(+) tumors were larger than MECA-79(-). Vascular invasion was also significantly frequent in MECA-79(+) tumors. In addition, Kaplan-Meier analysis showed that patients with MECA-79-positive lung adenocarcinoma had significantly worse recurrence-free survival.

However, this study includes several limitations, and its clinical significance appears limited. Therefore, the authors need to address several key issues to strengthen the manuscript.

Concerns:

#1. Regarding the patient selection process (Page 5, line 97-98. Page 15, line 273-276):

In this study, the inclusion criteria were a pathological diagnosis of primary invasive lung adenocarcinoma, pathological stage 1A. However, it is unclear why the authors selected only patients with stage 1A and adenocarcinoma. The clinical significance of limiting the study population to stage 1A lung adenocarcinoma should be clarified. If there are specific reasons for this selection, it should be stated explicitly in the manuscript.

#2. Regarding the statistical analysis (Page 7):

The descriptions of statistical analysis are insufficient. The authors should clearly specify which statistical tests were used to compare the two groups in the Methods section. In particular, it should be described which tests were applied for continuous variables and which were used for categorical variables.

#3. Regarding the small sample size.

In this study, the number of patients in the MECA-79(+) tumor group is only 10. Small sample sizes are prone to statistical instability and bias, which can lead to potentially unreliable results. Therefore, it may be worthwhile to consider including other histological types of lung cancer or more advanced stages to increase sample size and enhance the robustness of the findings.

Minor Points:

#1. (Page8: line184, Page10: Table 1)

The presentation of postoperative recurrence

→ The localization of postoperative recurrence

Reviewer #2: This is an automated report for PONE-D-25-16991. This report was solicited by the PLOS One editorial team and provided by ScreenIT.

ScreenIT is an independent group of scientists developing automated tools that analyze academic papers. A set of automated tools screened your submitted manuscript and provided the report below. Each tool was created by your academic colleagues with the goal of helping authors. The tools look for factors that are important for transparency, rigor and reproducibility, and we hope that the report might help you to improve reporting in your manuscript. Within the report you will find links to more information about the items that the tools check. These links include helpful papers, websites, or videos that explain why the item is important. While our screening tools aim to improve and maintain quality standards they may, on occasion, miss nuances specific to your study type or flag something incorrectly. Each tool has limitations that are described on the ScreenIT website. The tools screen the main file for the paper; they are not able to screen supplements stored in separate files. Please note that the Academic Editor had access to these comments while making a decision on your manuscript. The Academic Editor may ask that issues flagged in this report be addressed. If you would like to provide feedback on the ScreenIT tool, please email the team at ScreenIt@bih-charite.de. If you have questions or concerns about the review process, please contact the PLOS One office at plosone@plos.org.

Reviewer #3: This retrospective study evaluated 195 patients with pathological stage IA lung adenocarcinoma and found that MECA-79 expression in tumor cells (10 cases (5.1%)) was associated with larger invasive tumor size, higher vascular invasion, and increased 5-year recurrence. Kaplan-Meier analysis showed significantly worse recurrence-free survival in MECA-79–positive cases, and multivariate analysis confirmed MECA-79 as an independent predictor of recurrence. These findings suggest that tumoral MECA-79 expression may contribute to a metastasis-promoting microenvironment and warrants validation in larger cohorts.

The authors acknowledged the small number of MECA-79(+) tumors as a key limitation of the study. In Tables 2 and 3, univariable and multivariable logistic regression analyses were used to evaluate clinical and pathological predictors of postoperative recurrence within 5 years. However, given the time-to-event nature of recurrence and survival outcomes, a Cox regression analysis with hazard ratios would be more appropriate than odds ratios.

Additionally, due to the limited number of recurrence or death events, it may be more appropriate to present univariable analysis only. The multivariable analysis shown in Table 3, which includes only two variables, provides limited support for the conclusions and may not be statistically robust.

For Figure 3, please include the number of patients at risk beneath the survival curves to enhance clarity and interpretability.

Lastly, there is a discrepancy between the study period and number of patients described in the Figure 1 legend, compared to what is reported in the figure itself and the main text. Please revise to ensure consistency across the manuscript.

6. PLOS authors have the option to publish the peer review history of their article (what does this mean? ). If published, this will include your full peer review and any attached files.

**Do you want your identity to be public for this peer review?** For information about this choice, including consent withdrawal, please see our Privacy Policy .

Reviewer #1: No

Reviewer #2: No

Reviewer #3: No

---

## [Author Response · Author response to Decision Letter 1]

4 Aug 2025

AUTHORS’ RESPONSES TO COMMENTS FROM THE ACADEMIC EDITOR AND REVIEWERS

Re: PONE-D-25-16991　“Clinical and pathological implications of the presence of MECA-79-expressing tumor cells in pathological stage IA lung adenocarcinoma”

Academic Editor:

Thank you for submitting your manuscript to PLOS ONE. After careful consideration, we feel that it has merit but does not fully meet PLOS ONE’s publication criteria as it currently stands. Therefore, we invite you to submit a revised version of the manuscript that addresses the points raised during the review process.

The reviewers provided constructive feedback, particularly regarding the statistical analysis, including the use of univariable and multivariable models given the limited number of events. Please incorporate more rigorous statistical methods to avoid potential overfitting and strengthen the validity of the findings. Furthermore, integrating a bioinformatic analysis of publicly available datasets to explore the expression of genes involved in MECA-79 glycan biosynthesis could provide additional mechanistic insights and further support the clinical relevance of MECA-79 in tumor biology.

[Authors’ response]

Thank you for the constructive comments and valuable insights. We have revised our manuscript in response to the reviewers’ comments, as highlighted on the following pages. Moreover, we have revised the file names of figures, tables, and supporting information to comply with the journal’s requirements.

In response to the comments on our statistical methods from you and reviewer #3, we have excluded the multivariate logistic regression model including all candidate factors (Table 2 in the original version) from the manuscript to avoid overfitting.

While we acknowledge your concern regarding the limited number of events, our multivariate model with two selected variables represents a conservative approach that meets established statistical guidelines (i.e., the rule of ten events per variable). Although not optimal and providing limited support, this multivariate approach adjusts for confounding factors, which cannot be addressed using univariate analysis. Furthermore, performing Cox proportional hazards analysis while retaining the logistic regression models enables statistical triangulation, which potentially improves the reliability of the results.

Therefore, we have retained the multivariate logistic models including two variables in Table 3. In addition, we have conducted Cox proportional hazards regression analysis to evaluate the association of clinical and pathological characteristics with recurrence-free survival. We employed the same approach used in our logistic regression analysis by including only two candidate variables, including the presence of MECA-79+ tumor cells in the multivariate model, to follow the rule of ten events per variable (the number of recurrence or death events was 30).

We have revised the text in the Statistical analysis subsection of the Materials and Methods section as follows:

“Given the 17 postoperative recurrences observed within five postoperative years (as described below), two independent variables, including the presence of MECA-79+ tumor cells, were selected for multivariable model construction to avoid overfitting.” [Page 9, Lines 195–198 of Marked Copy]

“Additionally, univariate and multivariate Cox proportional hazards regression models were used to evaluate the association of clinical and pathological characteristics with the RFS of patients with LUAD. We employed the same approach as that used in the logistic regression analysis, with only two candidate variables, including the presence of MECA-79+ tumor cells in the multivariate model, to follow the rule of ten events per variable (the number of recurrence or death events was 30, as described below) [28].” [Page 9, Lines 201–207 of Marked Copy]

Additionally, we have revised Table 2 and added Tables 4 and 5. [Pages 12-13, 16 and 18 of Marked Copy]

Furthermore, we analyzed the human lung adenocarcinoma dataset from The Cancer Genome Atlas database using cBio Cancer Genomics Portal and Survival Genie 2.0. This investigation focused on the mRNA expression of genes associated with MECA-79 biosynthesis and their association with overall survival. The majority of LUAD showed upregulation of genes associated with MECA-79 biosynthesis.

We have added the Analysis using publicly available genomic database subsection to the Materials and Methods section as below:

“Analysis using publicly available genomic databases

For a comprehensive examination of both gene expression patterns and survival outcomes, robust, publicly available cancer genomics resources were utilized. The LUAD dataset from The Cancer Genome Atlas (TCGA), which includes 565 cases and is a widely used resource for cancer genomics, was accessed via two platforms: the cBio Cancer Genomics Portal (https://www.cbioportal.org/) [20] and Survival Genie 2.0 (https://bhasinlab.bmi.emory.edu/SurvivalGenie2/home) [21].

The cBio Cancer Genomics Portal was used to analyze gene expression in LUAD tumor cells. Specifically, we examined the expression levels of genes associated with MECA-79 antigen biosynthesis (e.g., B3GNT3, B4GALT4, CHST2, and CHST4) [22–25]. Gene expression levels in LUAD tumor cells were compared with those of normal cells within TCGA LUAD dataset. In addition, co-expression analysis was performed to identify genes showing significant correlation (Spearman’s correlation coefficient > 0.5) with the expression of B3GNT3 and B4GALT4, providing insight into potential functional associations.

Survival Genie 2.0 platform was employed to analyze overall survival (OS) in TCGA LUAD cohort. Patients were stratified into high- and low-expression groups for each MECA-79–associated gene based on the median expression value.” [Pages 7-8, Lines 160-177 of Marked Copy]

We have also revised the text in the Discussion section as follows:

“The MECA-79 biosynthetic pathway involves a series of glycosylation and sulfation steps that generate specialized O-glycan structures on proteins (Fig. 4A). Specifically, the enzymes beta-1,3-N-acetylglucosamine transferase 3 (B3GNT3) [22], carbohydrate sulfotransferase (CHST) family (CHST2 and CHST4)[23,24], and beta-1,4-galactosyltransferase 4 (B4GALT4)[25] have been shown to be essential for the production of MECA-79 antigen.

In an analysis of TCGA LUAD dataset (n = 565) using cBio Cancer Genomics Portal (https://www.cbioportal.org/) [Cerami E et al. Cancer Discov 2012], 87.5% of human LUAD samples showed gene alterations in B3GNT3, B4GALT4, CHST2, or CHST4, predominantly mRNA upregulation (78.6%), as shown for B3GNT3 and B4GALT4 in Fig. 4B. This, may contribute to tumoral expression of MECA-79 antigen in LUAD.” [Pages 19-20, Lines 308-318 of Marked Copy]

“Further analysis of TCGA LUAD datasset using Survival Genie 2.0 (https://bhasinlab.bmi.emory.edu/SurvivalGenie2/home) [21] indicated that high expression (above the median) of B3GNT3 and B4GALT4 was associated with diminished OS (P = 0.008 and 0.005, respectively, as shown in Fig. 5). Furthermore, overexpression of B3GNT3 protein is associated with diminished disease-free survival and OS with advanced TNM stage [21,29,30], indicating the association of abnormal glycosylation with malignant transformation in LUAD.” [Page 20, Lines 328-334 of Marked Copy]

“For example, analysis of TCGA LUAD dataset (n = 565) using the cBio Cancer Genomics Portal revealed that B3GNT3 is co-expressed with multiple genes associated with poor prognosis in patients with LUAD (Table 6) [33–42]. This finding suggests that MECA-79+ LUAD tumor cells may enhance their malignant potential” [Page 21, Lines 358-362 of Marked Copy]

Additionally, we have added Figures 4 and 5 and Table 6. [Pages 20, 20-21 and 22 of Marked Copy]

Reviewer #1:

It was a great pleasure to review the manuscript “Clinical and pathological implications of the presence of MECA-79-expressing tumor cells in pathological stage ⅠA lung adenocarcinoma” by Tomohito Saito, et al. In this study, the authors investigated the relationship between MECA-79(+) tumors and prognosis in 195 patients who underwent lobectomy for stage 1A lung adenocarcinoma. They revealed that the solid size and invasive size of MECA-79(+) tumors were larger than MECA-79(-). Vascular invasion was also significantly frequent in MECA-79(+) tumors. In addition, Kaplan-Meier analysis showed that patients with MECA-79-positive lung adenocarcinoma had significantly worse recurrence-free survival.

However, this study includes several limitations, and its clinical significance appears limited. Therefore, the authors need to address several key issues to strengthen the manuscript.

Concerns: #1.

Regarding the patient selection process (Page 5, line 97-98. Page 15, line 273-276): In this study, the inclusion criteria were a pathological diagnosis of primary invasive lung adenocarcinoma, pathological stage 1A. However, it is unclear why the authors selected only patients with stage 1A and adenocarcinoma. The clinical significance of limiting the study population to stage 1A lung adenocarcinoma should be clarified. If there are specific reasons for this selection, it should be stated explicitly in the manuscript.

[Authors’ response]

In the Introduction section of our manuscript, we highlighted the rate of postoperative recurrence for patients with pathological stage IA NSCLC is approximately 15%, which is equivalent to that of pathological stage II colorectal cancer. Our primary clinical question was why such a high proportion of patients with pathological stage IA NSCLC experience postoperative recurrence despite complete resection. The rationale for our inclusion criteria was as follows: We specifically analyzed pathological stage IA cases to minimize confounding variables associated with advanced disease stages. We also restricted our cohort to adenocarcinoma to eliminate potential confounding effects due to histological variations, given the known heterogeneity in biological behavior among NSCLC subtypes [Travis WD et al., J Thorac Oncol 2015]. Further, we included only patients who underwent complete (R0) resection with lobectomy to control for potential confounding effects related to surgical procedures or residual disease status.

In response to the reviewer’s comment, we have revised the Introduction section and Patient selection subsection in the Materials and Method section as below to further clarify the rationale for the inclusion criteria used in this study:

“Our primary clinical question is why such a high proportion of patients with pathological stage IA NSCLC experience postoperative recurrence despite complete resection. Delineation of the underlying mechanisms is essential for improving clinical outcomes.” [Page 3, Lines 63–66 of Marked Copy]

“The rationale for our inclusion criteria was as follows: We specifically analyzed pathological stage IA cases to minimize confounding variables associated with advanced disease stages. Furthermore, we restricted our cohort to adenocarcinoma to eliminate potential confounding effects due to histological variations, given the known heterogeneity in biological behavior among NSCLC subtypes [13]. We only included the patients who underwent complete (R0) resection with lobectomy to control for confounding effects related to surgical procedures or residual disease status.” [Page 5, Lines 105–111 of Marked Copy]

We have also cited an additional paper (Reference #13, Travis WD et al., J Thorac Oncol 2015) and updated the reference numbers accordingly.

Reviewer #1:

Concerns: #2. Regarding the statistical analysis (Page 7):

The descriptions of statistical analysis are insufficient. The authors should clearly specify which statistical tests were used to compare the two groups in the Methods section. In particular, it should be described which tests were applied for continuous variables and which were used for categorical variables.

[Authors’ response]

In response to the reviewer’s comment, we have revised the Statistical analysis subsection in the Materials and Methods section as below to further clarify which statistical test was used to compare the two groups:

“The Kruskal–Wallis test was used to analyze continuous variables, while Fisher’s exact test was used for categorical variables in comparing two groups.” [Page 8, Lines 180–181 of Marked Copy]

Reviewer #1:

Concerns #3. Regarding the small sample size.

In this study, the number of patients in the MECA-79(+) tumor group is only 10. Small sample sizes are prone to statistical instability and bias, which can lead to potentially unreliable results. Therefore, it may be worthwhile to consider including other histological types of lung cancer or more advanced stages to increase sample size and enhance the robustness of the findings.

[Authors’ response]

Our aim was to investigate the risk of postoperative recurrence in patients with pStage IA LUAD. Thus, inclusion of other histological types or advanced stages are beyond the scope of this study. We agree, however, that other histological types or advanced stages of cancer should be investigated in future research.

In response to the reviewer’s comment, we have revised the Discussion section as follows:

“Thus, the clinical and pathological significance of the presence of MECA-79+ tumor cells in patients with more advanced pathological stages (i.e., IB–IV) or with other histological types, such as non-adenocarcinoma NSCLC (e.g., squamous cell carcinoma) should be investigated in future studies.” [Page 23, Lines 372–376 of Marked Copy]

Reviewer #1:

Minor Points: #1. (Page8: line184, Page10: Table 1)

The presentation of postoperative recurrence → The localization of postoperative recurrence

[Authors’ response]

In response to the reviewer’s comment, we have revised description in Table 1 as follows: “Localization of postoperative recurrence” [Page 11 of Marked Copy]

Additionally, we have revised the related description in the Results section as follows:

“The localization of postoperative recurrence was not significantly different between the two study groups (P = 0.77).” [Page 10, Lines 221 of Marked Copy]

Reviewer #2:

This is an automated report for PONE-D-25-16991. This report was solicited by the PLOS One editorial team and provided by ScreenIT.

ScreenIT is an independent group of scientists developing automated tools that analyze academic papers. A set of automated tools screened your submitted manuscript and provided the report below. Each tool was created by your academic colleagues with the goal of helping authors. The tools look for factors that are important for transparency, rigor and reproducibility, and we hope that the report might help you to improve reporting in your manuscript. Within the report you will find links to more information about the items that the tools check. These links include helpful papers, websites, or videos that explain why the item is important. While our screening tools aim to improve and maintain quality standards they may, on occasion, miss nuances specific to your study type or flag something incorrectly. Each tool has limitations that are described on the ScreenIT website. The tools screen the main file for the paper; they are not able to screen supplements stored in separate files. Please note that the Academic Editor had access to these comments while making a decision on your manuscript. The Academic Editor may ask that issues flagged in this report be addressed. If you would like to provide feedback on the ScreenIT tool, please email the team at ScreenIt@bih-charite.de. If you have questions or concerns about the review process, please contact the PLOS One office at plosone@plos.org.

[Authors’ response]

We thank the editorial team for soliciting the report.

Reviewer #3:

This retrospective study evaluated 195 patients with pathological stage IA lung adenocarcinoma and found that MECA-79 expression in tumor cells (10 cases (5.1%)) was associated with larger invasive tumor size, higher vascular invasion, and increased 5-year recurrence. Kaplan-Meier analysis showed significantly worse recurrence-free survival in MECA-79–positive cases, and multivariate analysis confirmed MECA-

---

## [Decision Letter · Decision Letter 1]

27 Aug 2025

PONE-D-25-16991R1Clinical and pathological implications of the presence of MECA-79-expressing tumor cells in pathological stage IA lung adenocarcinomaPLOS ONE

Dear Dr. Saito,

Thank you for submitting your manuscript to PLOS ONE. After careful consideration, we feel that it has merit but does not fully meet PLOS ONE’s publication criteria as it currently stands. Therefore, we invite you to submit a revised version of the manuscript that addresses the points raised during the review process.

Although your manuscript has undergone substantial revision, the reviewer has identified remaining statistical and bioinformatic concerns. We kindly request that you address these issues in your revised submission.

We look forward to receiving your revised manuscript.

Kind regards,

Hyun-Sung Lee, M.D., Ph.D.

Academic Editor

PLOS ONE

Journal Requirements:

Reviewers' comments:

Reviewer's Responses to Questions

**Comments to the Author**

1. If the authors have adequately addressed your comments raised in a previous round of review and you feel that this manuscript is now acceptable for publication, you may indicate that here to bypass the “Comments to the Author” section, enter your conflict of interest statement in the “Confidential to Editor” section, and submit your "Accept" recommendation.

Reviewer #1: All comments have been addressed

Reviewer #3: (No Response)

2. Is the manuscript technically sound, and do the data support the conclusions?

Reviewer #1: Yes

Reviewer #3: Partly

3. Has the statistical analysis been performed appropriately and rigorously? 

Reviewer #1: Yes

Reviewer #3: No

4. Have the authors made all data underlying the findings in their manuscript fully available?

Reviewer #1: Yes

Reviewer #3: Yes

5. Is the manuscript presented in an intelligible fashion and written in standard English?

Reviewer #1: Yes

Reviewer #3: Yes

6. Review Comments to the Author

Reviewer #1: (No Response)

Reviewer #3: Saito et al. investigated the prognostic significance of ectopic MECA-79 expression in pathological stage IA lung adenocarcinoma. While the topic is clinically relevant, the revised manuscript still contains fundamental flaws that undermine its conclusions due to statistical fragility from severe sample imbalance, inconsistent analyses, and contradictory TCGA data.

Here are the concerns that need to be addressed:

1) The MECA-79 (+) group is too small (n=10), making the study susceptible to Type I errors or anecdotal results. The “rule of ten” cited by authors prevents convergence issues but does not guarantee against overfitting with such an imbalance (10 vs. 185 patients). This fragility is clearly demonstrated in their own Cox proportional hazards regression analysis. MECA-79’s prognostic significance disappears (p=0.067) when combined with vascular invasion, suggesting collinearity between MECA-79 expression and vascular invasion, rather than independent prediction. Furthermore, the author's use of “statistical triangulation” through both logistic and Cox regression is unconvincing, as different model and variable combinations lead to inconsistent conclusions. This inconsistency reveals the instability of the findings rather than their reliability.

2) The authors added a bioinformatic analysis using the TCGA database. However, this analysis critically weakens their central hypothesis. The authors report that 78.6% of LUAD samples in the TCGA showed mRNA upregulation of genes related to MECA-79 biosynthesis. In contrast, their own analysis found that only 5.1% of patients (10 of 195) showed actual MECA-79 epitope expression. This major discrepancy between high mRNA expression and low protein/epitope presence makes TCGA data inappropriate as supportive evidence.

3) The authors limited their cohort to pStage IA LUAD to “minimizing confounding variables.” While this is a reasonable approach, they use survival data from the TCGA cohort, which includes relatively advanced stages, to support their claims about the prognostic value of MECA-79-related genes. This is methodologically inconsistent.

4) The manuscript’s introduction compares the 15% recurrence rate of pStage IA NSCLC to that of pStage II colorectal cancer. This comparison is clinically irrelevant and should be removed. TNM staging systems are uniquely calibrated for each cancer type based on its specific anatomical and biological behavior. A direct comparison of survival rates between different stages of different cancers is meaningless.

5) In their response to Reviewer #1, the authors state, “The Kruskal-Wallis test was used to… in comparing two groups.” The Kruskal-Wallis test is a non-parametric method for comparing three or more groups. For a two-group comparison, the Mann-Whitney U test is appropriate. This fundamental error in reporting the statistical methods further erodes confidence in the overall statistical rigor of the study.

6) Given that recurrence and survival represent time-to-event outcomes, logistic regression analysis is less appropriate than the Cox regression model. However, considering the severely limited sample size, the authors should focus on the time-to-event analysis with univariable analysis only to avoid overfitting and ensure statistical validity.

7. PLOS authors have the option to publish the peer review history of their article (what does this mean? ). If published, this will include your full peer review and any attached files.

**Do you want your identity to be public for this peer review?** For information about this choice, including consent withdrawal, please see our Privacy Policy .

Reviewer #1: No

Reviewer #3: No

---

## [Author Response · Author response to Decision Letter 2]

1 Oct 2025

AUTHORS’ RESPONSES TO COMMENTS FROM THE ACADEMIC EDITOR AND REVIEWERS

PONE-D-25-16991R1 “Clinical and pathological implications of the presence of MECA-79-expressing tumor cells inpathological stage IA lung adenocarcinoma”

Reviewer #3:

Saito et al. investigated the prognostic significance of ectopic MECA-79 expression inpathological stage IA lung adenocarcinoma. While the topic is clinically relevant, the revisedmanuscript still contains fundamental flaws that undermine its conclusions due to statistical fragility from severe sample imbalance, inconsistent analyses, and contradictory TCGA data.

Here are the concerns that need to be addressed:

1) The MECA-79+ group is too small (n=10), making the study susceptible to Type I errors or anecdotal results. The “rule of ten” cited by authors prevents convergence issues but does not guarantee against overfitting with such an imbalance (10 vs. 185 patients). This fragility is clearly demonstrated in their own Cox proportional hazards regression analysis. MECA-79’s prognostic significance disappears (p=0.067) when combined with vascular invasion, suggesting collinearity between MECA-79 expression and vascular invasion, rather than independent prediction. Furthermore, the author's author’s use of “statistical triangulation” through both logistic and Cox regression is unconvincing, as different model and variable combinations lead to inconsistent conclusions. This inconsistency reveals the instability of the findings rather than their reliability.

[Authors’ response]

We appreciate the reviewer’s thoughtful critique. We agree that the MECA-79+ group is small and that this imbalance limits the robustness of our conclusions. We therefore present univariable Cox regression as the primary analysis in the main text and have moved the multivariable Cox models to the Supplementary Materials and have removed logistic models to avoid overstating fragile results. We further acknowledge the statistical fragility in the Limitations.

In response to the Reviewer’s comment, we have revised the Limitations of the study as follows:

“First, our results are based on a retrospective single-institutional study with a small sample size, particularly the MECA-79⁺ group (n = 10). This imbalance increases statistical fragility, limits power, and raises the possibility of collinearity with vascular invasion, which may confound the apparent prognostic effect of MECA-79. Therefore, the present study should be regarded as exploratory, and validation in a prospective, multi-institutional study with a larger sample size—ideally including stratification by vascular invasion—is required.”　 [Page 16, Lines 310–317 of Marked Copy]

2) The authors added a bioinformatic analysis using the TCGA database. However, this analysis critically weakens their central hypothesis. The authors report that 78.6% of LUAD samples in the TCGA showed mRNA upregulation of genes related to MECA-79 biosynthesis. In contrast, their own analysis found that only 5.1% of patients (10 of 195) showed actual MECA-79 epitope expression. This major discrepancy between high mRNA expression and low protein/epitope presence makes TCGA data inappropriate as supportive evidence.

[Authors’ response]

We thank the reviewer for this important observation. We agree that TCGA analysis was initially added in response to the Associate Editor’s suggestion, does not provide supportive evidence for the prognostic value of MECA-79. On the contrary, the discrepancy between frequent upregulation of biosynthetic enzymes in TCGA (78.6%) and the rare detection of MECA-79 epitopes in our cohort (5.1%) highlights a limitation of our study and reflects a broader challenge in glycobiology. As the reviewer correctly noted, this discordance weakens the hypothesis that transcriptional upregulation alone can predict MECA-79 epitope expression, because carrier proteins are also required for the display of the glycan epitop. We have now clarified in the Discussion section that this discrepancy reflects a well-recognized phenomenon in glycobiology: mRNA levels of glycosylation enzymes do not necessarily correspond to the presence of glycan epitopes, due to post-transcriptional regulation and the requirement of carrier proteins (Neelamegham et al., Glycobiology 2016; Bennun et al., PLOS Comput Biol 2013). Accordingly, we explicitly state that TCGA findings should be regarded only as contextual mechanistic background rather than prognostic evidence.

In response to the reviewer’s comment, we have revised the Discussion sections as follows:

“While such upregulation may provide a biological background for MECA-79 biosynthesis, MECA-79 epitope expression was rare (5.1%) in our cohort, despite frequent mRNA upregulation in TCGA. This discrepancy highlights a well-recognized phenomenon in glycobiology: mRNA levels of glycosylation enzymes do not necessarily correspond to the presence of glycan epitopes due to post-transcriptional regulation and the requirement of carrier proteins [25,26]. These findings suggest that TCGA data should be regarded as contextual mechanistic background rather than prognostic evidence. Nevertheless, future studies incorporating glycoproteomic profiling are needed to resolve this gap.”　 [Page 14, Lines 268–277 of Marked Copy]

3) The authors limited their cohort to pStage IA LUAD to “minimizing confounding variables.” While this is a reasonable approach, they use survival data from the TCGA cohort, which includes relatively advanced stages, to support their claims about the prognostic value of MECA-79-related genes. This is methodologically inconsistent.

[Authors’ response]

We agree with the reviewer that using TCGA survival data, which includes more advanced stages, was methodologically inconsistent with our focus on pStage IA LUAD. Accordingly, we have removed the description linking MECA-79–related enzyme mRNA expression with prognosis, and retained only the discussion focusing on the mechanistic background of MECA-79 expression. Accodingly, we have renamed Table 6 in the original version as Table 3 in the revised version.

In the revised Materials and Methods section, we clarify that the purpose of accessing the TCGA LUAD dataset was to explore the mechanistic background of tumoral MECA-79 expression, rather than its survival implications. The revised text reads as follows:

“For a comprehensive examination of gene expression patterns focusing on the mechanistic background of tumoral MECA-79 expression, a robust, publicly available cancer genomics resource was used. The LUAD dataset from The Cancer Genome Atlas (TCGA), which includes 565 cases and is a widely used resource for cancer genomics, was accessed via the cBio Cancer Genomics Portal (https://www.cbioportal.org/) [19].” [Page 16, Lines 310–317 of Marked Copy]

Additionally, we have removed Figure 5 and all related descriptions concerning survival implications from TCGA dataset.

We have also revised the limitations of the study as follows:

“Fourth, TCGA analysis revealed frequent upregulation of biosynthetic enzymes despite rare MECA-79 epitope expression in our cohort. This discrepancy reflects a common disconnect between transcript levels and glycan presentation, underscoring that TCGA findings cannot be considered supportive prognostic evidence but only contextual background.” [Page 17, Lines 325–329 of Marked Copy]

4) The manuscript’s introduction compares the 15% recurrence rate of pStage IA NSCLC to that of pStage II colorectal cancer. This comparison is clinically irrelevant and should be removed. TNM staging systems are uniquely calibrated for each cancer type based on its specific anatomical and biological behavior. A direct comparison of survival rates between different stages of different cancers is meaningless.

[Authors’ response]

We appreciate the reviewer’s comment and agree that comparing recurrence rates across different cancer types is clinically inappropriate. Accordingly, we have removed the cross-cancer comparison from the Introduction. In response to the reviewer’s comment, we deleted the sentence comparing the recurrence rate of pStage IA NSCLC with that of pStage II colorectal cancer.

5) In their response to Reviewer #1, the authors state, “The Kruskal-Wallis test was used to… in comparing two groups.” The Kruskal-Wallis test is a non-parametric method for comparing three or more groups. For a two-group comparison, the Mann-Whitney U test is appropriate. This fundamental error in reporting the statistical methods further erodes confidence in the overall statistical rigor of the study.

[Authors’ response]

We sincerely apologize for this error. We have corrected the text to report the Mann–Whitney U test for two-group comparisons. The numerical results remain unchanged.

In response to the reviewer’s comment, we have revised the Materials and Methods section as follows:

“The Mann–Whitney U test was used to analyze continuous variables, while Fisher’s exact test was used for categorical variables in comparing two groups.” [Page 8, Lines 169–170 of Marked Copy]

6) Given that recurrence and survival represent time-to-event outcomes, logistic regression analysis is less appropriate than the Cox regression model. However, considering the severely limited sample size, the authors should focus on the time-to-event analysis with univariable analysis only to avoid overfitting and ensure statistical validity.

[Authors’ response]

We thank the reviewer for this comment and agree that Cox regression is the appropriate method for time-to-event outcomes such as recurrence and survival. As noted above, we now present univariable Cox regression as the primary survival analysis. Multivariable Cox regression including is retained only in the Supplementary Material as an exploratory analysis to maintain transparency while avoiding overfitting in the main text, whereas logistic regression analyses have been removed.

We have revised the Materials and Methods section as follows:

“Univariable Cox proportional hazards regression was primarily used to evaluate the association of clinical and pathological characteristics with recurrence-free survival (RFS) in patients with LUAD. Significant factors (P < 0.05) in the univariable analysis were considered candidate risk factors. To minimize overfitting given the limited number of recurrence or death events (n = 3), multivariable Cox models –restricted to two candidate variables, including the presence of MECA-79+ tumor cells–were prespecified as exploratory analyses and are presented in the Supplementary Material (S1 Table).” [Page 8, Lines 181–188 of Marked Copy]

Accordingly, Tables 2 and 3, which presented the logistic regression analyses, have been removed. Table 4 in the original version, which presented the univariable Cox regression analysis, has been retained and renumbered as Table 2 in the revised version. Table 5 in the original version has been moved to the Supplementary Material as S1 Table. In addition, we revised the Results section to clarify that the multivariable Cox models are exploratory analyses, as follows:

“Additionally, results of exploratory multivariable Cox proportional hazards regression analyses are provided in the Supplementary Material (S1 Table).” [Page 12, Lines 240–241 of Marked Copy]

We have reported the results of exploratory multivariable Cox analyses in the Supplementary Material as follows:

“In exploratory analyses, the presence of MECA-79+ tumor cells showed a tendency toward association with worse RFS in some multivariable Cox proportional hazards regression models; however, this association was not consistent and disappeared in the model incorporating vascular invasion.” [Page 2, Lines 6-8 of Supplementary Material]

---

## [Editor Report · Decision Letter 2]

6 Oct 2025

Clinical and pathological implications of the presence of MECA-79-expressing tumor cells in pathological stage IA lung adenocarcinoma

PONE-D-25-16991R2

Dear Dr. Saito,

We’re pleased to inform you that your manuscript has been judged scientifically suitable for publication and will be formally accepted for publication once it meets all outstanding technical requirements.

Kind regards,

Hyun-Sung Lee, M.D., Ph.D.

Academic Editor

PLOS ONE
---

## [Editor Report · Acceptance letter]

PONE-D-25-16991R2

PLOS ONE

Dear Dr. Saito,

I'm pleased to inform you that your manuscript has been deemed suitable for publication in PLOS ONE. Congratulations! Your manuscript is now being handed over to our production team.

Kind regards,

on behalf of

Dr. Hyun-Sung Lee

Academic Editor

PLOS ONE